# Metabolic Dysfunction-Associated Steatotic Liver Disease in a Dish: Human Precision-Cut Liver Slices as a Platform for Drug Screening and Interventions

**DOI:** 10.3390/nu16050626

**Published:** 2024-02-23

**Authors:** Mei Li, Frederik T. Larsen, Marius C. van den Heuvel, Konstanze Gier, Alan R. Gorter, Dorenda Oosterhuis, Johan Bijzet, Vincent E. de Meijer, Kim Ravnskjaer, Anika Nagelkerke, Peter Olinga

**Affiliations:** 1Department of Pharmaceutical Technology and Biopharmacy, Groningen Research Institute of Pharmacy, University of Groningen, 9713 AV Groningen, The Netherlands; mei.li@rug.nl (M.L.); k.gier@rug.nl (K.G.); a.r.gorter@rug.nl (A.R.G.); d.oosterhuis@rug.nl (D.O.); 2Department of Biochemistry and Molecular Biology, University of Southern Denmark, 5230 Odense M, Denmark; tibert@bmb.sdu.dk (F.T.L.); ravnskjaer@bmb.sdu.dk (K.R.); 3Department of Pathology and Medical Biology, Pathology Section, University Medical Center Groningen, 9700 RB Groningen, The Netherlands; m.c.van.den.heuvel@umcg.nl; 4Amyloidosis Center of Expertise, University Medical Center Groningen, 9713 GZ Groningen, The Netherlands; j.bijzet@umcg.nl; 5Department of Laboratory Medicine, Division of Medical Immunology, University Medical Center Groningen, 9713 GZ Groningen, The Netherlands; 6Department of Surgery, Division of Hepato-Pancreato-Biliary Surgery and Liver Transplantation, University Medical Center Groningen, 9713 AV Groningen, The Netherlands; v.e.de.meijer@umcg.nl; 7Department of Pharmaceutical Analysis, Groningen Research Institute of Pharmacy, University of Groningen, 9713 AV Groningen, The Netherlands

**Keywords:** precision-cut liver slices (PCLSs), long-term incubation, metabolic dysfunction-associated steatotic liver disease (MASLD), non-alcoholic fatty liver disease (NAFLD), hepatic steatosis, liver fibrosis

## Abstract

Metabolic dysfunction-associated steatotic liver disease (MASLD) is a growing healthcare problem with limited therapeutic options. Progress in this field depends on the availability of reliable preclinical models. Human precision-cut liver slices (PCLSs) have been employed to replicate the initiation of MASLD, but a comprehensive investigation into MASLD progression is still missing. This study aimed to extend the current incubation time of human PCLSs to examine different stages in MASLD. Healthy human PCLSs were cultured for up to 96 h in a medium enriched with high sugar, high insulin, and high fatty acids to induce MASLD. PCLSs displayed hepatic steatosis, characterized by accumulated intracellular fat. The development of hepatic steatosis appeared to involve a time-dependent impact on lipid metabolism, with an initial increase in fatty acid uptake and storage, and a subsequent down-regulation of lipid oxidation and secretion. PCLSs also demonstrated liver inflammation, including increased pro-inflammatory gene expression and cytokine production. Additionally, liver fibrosis was also observed through the elevated production of pro-collagen 1a1 and tissue inhibitor of metalloproteinase-1 (TIMP1). RNA sequencing showed that the tumor necrosis factor alpha (TNFα) signaling pathway and transforming growth factor beta (TGFβ) signaling pathway were consistently activated, potentially contributing to the development of inflammation and fibrosis. In conclusion, the prolonged incubation of human PCLSs can establish a robust ex vivo model for MASLD, facilitating the identification and evaluation of potential therapeutic interventions.

## 1. Introduction

Metabolic dysfunction-associated steatotic liver disease (MASLD), previously termed non-alcoholic fatty liver disease (NAFLD), is a chronic and progressively developing condition impacting approximately 38% of the global population [1]. It is closely linked to metabolic syndromes, such as obesity and type 2 diabetes mellitus. MASLD results from fat accumulation within the liver and encompasses a spectrum of disease stages, ranging from an isolated lipid buildup or steatosis (metabolic dysfunction-associated steatotic liver, MASL), to its more active inflammatory manifestation, metabolic dysfunction-associated steatohepatitis (MASH), formerly known as nonalcoholic steatohepatitis (NASH). The persistence of MASH can ultimately lead to the development of fibrosis, cirrhosis, and even liver cancer. Although several drugs are in advanced stages of development, there is currently no approved cure for MASLD [2]. Long-term lifestyle modification with a focus on healthy diet, weight loss, and regular exercise remain the cornerstone of therapy in children and adults [3], yet willpower and persistence are required. So far, a major barrier to the development of therapies for MASLD is the lack of preclinical models of disease that are appropriately validated to represent the biology and outcomes of human liver diseases [4]. Though many advances in the development of preclinical models for MASLD have been made that have provided valuable insights on disease pathogenesis, only a few models recapitulate the key elements needed to be representative for human liver disease [4].

Therefore, the ideal MASLD models should mimic human disease as closely as possible if the objective is to enhance the likelihood that a drug that improves MASLD in the preclinical models also improves the disease in humans [4]. A relatively simple model, monolayer cultures of primary human hepatocytes or liver-based cell lines, fails to capture the interactions with other cell types seen in the liver [5]. Co-cultures of hepatocytes and non-parenchymal cells like Kupffer cells or fibroblasts and sandwich cultures of primary hepatocytes between two layers of collagen type 1 have been developed as well, but these planar culture systems have failed in representing the complex architecture of hepatic tissue in vivo. Therefore, here, we aimed to create an ex vivo model utilizing PCLSs which preserve cell–cell contacts and the extracellular matrix (ECM). This model serves as a valuable tool for investigating liver diseases, given its capacity to closely mimic the in vivo context [6].

To date, several studies have utilized PCLSs for the research of MASLD. Healthy murine PCLSs exposed to steatotic conditions, such as high concentration of fructose, glucose, insulin and/or fatty acids for a maximum of 48 h incubation could develop the phenotype of MASLD, represented by the increased positive area of Oil Red O staining and triglyceride (TG) content [7,8]. Still, these organotypic models did not instigate an inflammatory response [8] or only showed a modest increase in *Il1β* expression [9]. Similar treatments of rat [10] and human [11,12] PCLSs also led to lipid deposition, TG secretion, and/or lipotoxicity. However, these PCLS models either show no evidence of MASH or a late stage of MASLD. In this study, we prolonged the incubation time of healthy human PCLSs beyond the previously reported 48 h, under conditions that induce steatotic liver disease. We investigated the progression of MASLD including steatosis, steatohepatitis, and liver fibrosis. The aim of this study was to ascertain whether an extended incubation period could lead to the advancement of MASLD beyond the initial stage of simple hepatic steatosis.

## 2. Materials and Methods

### 2.1. Collection of Human Liver Tissue and Preparation of PCLSs

The use of human material was approved by the Medical Ethical Committee of the University Medical Centre Groningen (UMCG) according to Dutch legislation and the Code of Conduct for dealing responsibly with human tissue in the context of health research (www.federa.org), refraining the need of written consent for “further use” of coded-anonymous human tissue. We collected tissues and prepared PCLSs as previously published [13]. Briefly, human liver tissues were sourced from individuals undergoing partial hepatectomy or from organ donors. These tissues were conserved in a University of Wisconsin (UW) preservation solution (Cat. BUWC, Bridge to Life Ltd., London, UK) at 4 °C. Biopsies with a 6 mm diameter were sectioned and subsequently sliced with a Krumdieck tissue slicer (Alabama Research and Development, Munford, TN, USA), which was filled with an ice-cold Krebs–Henselheit buffer, following the protocol described by de Graaf et al. [14]. PCLSs were estimated to be approximately 250 μm in thickness and were stored at 4 °C in UW preservation solution. The number of PCLSs and analyses conducted were constrained by the size of tissue available from donations. For each analytical procedure, tissues from a minimum of five distinct donor livers were utilized. Livers that failed to exhibit adenosine triphosphate (ATP) production after 1 h of culture were considered non-viable and consequently excluded from the study.

### 2.2. PCLS Culture and Induction of Steatosis

PCLSs were cultured in Williams medium E with GlutaMAX™ (Cat. 32551020, Invitrogen, Bleiswijk, The Netherlands), supplemented with gentamycin (50 µg/mL; Cat. 15750-037, Invitrogen) and glucose (25 mM; Cat. 1.08342.1000, Merck, Darmstadt, Germany), hereinafter referred to as “WEGG”. To induce steatosis, additional glucose (36 mM), fructose (5 mM, Cat. F3510-100G, Sigma-Aldrich, St. Louis, MO, USA), human insulin (1 nM, Cat. I9278, Sigma-Aldrich), palmitic acid (240 µM, Cat. P0500-10G, Sigma-Aldrich), and oleic acid (480 µM, Cat. 75096-1L, Sigma-Aldrich) were added, a combination denoted as “GFIPO”. Palmitic acid and oleic acid were solubilized in 0.04% bovine serum albumin (BSA; Cat. A2153-100G, Sigma-Aldrich). The fatty acids were initially dissolved in 0.1 M sodium hydroxide (Cat. 1.06469.5000, Merck) at 70 °C, followed by combination with pre-warmed BSA in water at 55 °C. It was observed that the final concentrations of BSA and sodium hydroxide did not influence the pH of the medium nor affect the PCLSs’ viability. The medium preparation and final concentrations were based on previous experiments [10]. PCLSs were incubated at 37 °C in an atmosphere containing 20% O_2_ and 5% CO_2_ for periods ranging from 24 to 96 h. The medium was renewed at 24 h intervals, and samples were collected at the same frequency.

### 2.3. ATP Measurement

Metabolic activity and viability of PCLSs were assessed by measuring ATP content using the ATP Bioluminescence Assay Kit CLS II (Cat. 11699695001, Roche Diagnostics, Mannheim, Germany), as previously described [14]. ATP was measured per slice and corrected for total protein content.

### 2.4. Triglyceride Quantitation

The triglyceride (TG) levels in PCLSs and the medium were quantified utilizing the Trig/GB kit (Cat. 11877771, Roche Diagnostics), according to the protocol provided by the manufacturer. In short, PCLSs were snap-frozen and preserved at −80 °C pending analysis. They were subsequently homogenized in a Tris-buffered saline solution, and lipids were extracted using the Bligh and Dyer method [15]. Media samples, pooled from 12 PCLSs derived from the same donor and identical conditions, were stored at −20 °C until further processing. The TG values were determined by measuring the absorption at 540 nm after 1 h and adjusting for the total protein content [10].

### 2.5. Total Protein Measurement

The total protein in the PCLSs was assessed via the Lowry assay (Cat. 5000113, Cat. 5000114; BioRad DC Protein Assay, Hercules, CA, USA) and a BSA calibration curve [16]. The quantified protein concentrations were used to normalize the ATP and TG measurements as previously published [10,13].

### 2.6. RNA Sequencing and Quantitative Real-Time PCR

Transcriptomic alterations were assessed through total RNA sequencing and quantitative Real-Time PCR (RT-qPCR). For each experimental condition, three to nine PCLSs from a single donor were combined, snap-frozen in liquid nitrogen with Qiazol buffer (Cat. No. 79306, Qiagen, Venlo, The Netherlands) added, and, subsequently, stored at −80 °C pending further analysis. For homogenization and lysis, PCLSs were processed using Qiazol buffer, followed by vigorous mixing with one-fifth the volume of chloroform (Cat. 1.02445.4000, Sigma-Aldrich). Phase separation was achieved using extraction tubes, and RNA was isolated via the RNeasy Lipid Tissue Mini Kit (Ref. 74106, Qiagen) [10,13]. The integrity and concentration of the extracted RNA were determined using the RNA Screen Tape assay on the TapeStation 4200 System (Agilent, Santa Clara, CA, USA). 

For next-generation sequencing (NGS), the construction of libraries was performed using the NEBNext Ultra RNA Library Prep Kit for Illumina (New England Biolabs, San Diego, CA, USA) according to the manufacturer’s protocol. RNA was paired-end sequenced using the NovaSeq™ 6000 platform (Illumina, San Diego, CA, USA). Reads were aligned with STAR (v. 2.7.3a) [17] to the human genome assembly (GRCh38, Ensembl release 101). FeatureCounts (v.2.0) [18] was employed for exon read quantification. The quality of raw sequencing was assessed using FastQC (v.0.11.8) [19] and MultiQC (v.1.10.1) [20]. NGS was performed on seven liver donors per condition.

DESeq2 (v. 1.40.2) [21] was used to identify differentially expressed genes (DEGs) by fitting a negative binominal model to the raw counts. A Wald test with α-error accumulation correction using Benjamini–Hochberg implemented in the DESeq2 package was employed to determine genes as differentially expressed. Adjusted *p*-values < 0.05 were considered statistically significant. Regularized log transformation was used to normalize raw gene counts for further analysis. The regulation of biological processes was assessed through a gene set enrichment analysis (GSEA) using GSEA software (v. 4.3.2) [22,23]. 

The expression levels of critical genes associated with inflammation and fibrosis were quantified using RT-qPCR as previously published [10,13]. The process of the reverse transcription of RNA into complementary DNA (cDNA) was facilitated using the Reverse Transcription System (Promega, Leiden, The Netherlands). Subsequent RT-qPCR analysis was conducted on a ViiA 7 Real-Time PCR System (Applied Biosystems, Waltham, MA, USA), utilizing SYBR Green primers (Appendix A) and SYBR Green reagent (Cat. No. 04913914001, Roche Diagnostics). The threshold cycle (Ct) values were normalized to the Ct values of reference gene *YWHAZ* (resulting in ΔCt values) and further compared to those of the control group to calculate the relative expression levels (ΔΔCt). The results were presented as the mean fold change in expression (2^−ΔΔCt^), computed using the method delineated by Livak et al. [24]. Statistical analyses were subsequently performed using the ΔΔCt values.

### 2.7. Measurement of Excreted Proteins

Mediums from 12 PCLSs per condition (one 12-well plate) were pooled together and stored at −20 °C. Cytokines and chemokines were measured with Luminex^®^ Multiplex Assay (LXSAHM-13, R&D Systems, Minneapolis, MN, USA) according to the manufacturer’s instructions.

Pro-collagen Ia1 concentrations were quantitatively assessed in a 1:20 dilution of a pooled culture medium utilizing the Human Pro-Collagen I alpha 1 ELISA Kit (ab210966, Abcam, Cambridge, UK), following the manufacturer’s protocol. The data are expressed as relative values compared to the control (WEGG, 24 h) [10,13].

ApoB100 concentration was measured in a 1:1 dilution using a Human Apolipoprotein B ELISA development kit (3715-1H-6, Mabtech, Nacka Strand, Sweden) according to instructions from the manufacturer.

### 2.8. Histology

PCLSs were fixed in 4% buffered formalin overnight and stored at 4 °C in 70% ethanol. Fixed PCLSs were embedded in paraffin and sectioned 4 μm thick. Tissue morphology and fibrosis were assessed through hematoxylin and eosin (H&E) staining (Hematoxyline: Cat. 4085.9002, Klinipath, Duiven, The Netherlands; Eosin: Cat. HT110232, Sigma-Aldrich) and Picrosirius Red (PSR) staining (ab150681, Abcam) according to the standard histological procedure. Stained tissue sections were scanned using a Nanozoomer Digital Pathology Scanner (NDP Scan U10074-01, Hamamatsu Photonics K.K., Shizuoka, Japan) [10,13]. ImageJ was used to quantify the stained areas. The histopathologic percentages of necrosis and fat vesicles were determined by an experienced pathologist (MH).

### 2.9. Statistics

Statistical significance was identified by comparing each experimental condition to its respective control. Replicates included at least five distinct livers, with a minimum of three precision-cut liver slices (PCLSs) per condition sourced from each liver. The data are presented as mean ± standard error of the mean (SEM). Statistical analyses were conducted using GraphPad Prism version 9.2.0 (GraphPad Software Inc., Boston, MA, USA). Group comparisons were made using one-way ANOVA followed by Dunnett’s multiple comparisons test, in addition to paired two-tailed *t*-tests. A *p*-value less than 0.05 was considered indicative of statistical significance [10,13].

## 3. Results

### 3.1. Characterization of PCLSs

#### 3.1.1. Slices Retain Viability after Incubation

Slices were incubated in the GFIPO medium to induce the MASLD phenotype and in the WEGG medium as the control. Notably, slices were incubated in 20% O_2_ as they exhibited stable viability up to 96 h compared to 80% O_2_, and three out of four livers showed down-regulation in ATP production at 80% O_2_ compared to 20% O_2_ (Appendix A). The analysis of the PCLS morphology using H&E staining revealed that the slices showed no obvious signs of the loss of cell viability (Figure 1A). However, over time, increased necrotic areas were observed in PCLSs after incubation in both WEGG and GFIPO, as compared to the 24 h incubation in WEGG (Figure 1B). This indicates that incubation could cause moderate cell death towards the end, regardless of the medium conditions used. Measurements of ATP showed that GFIPO resulted in lower ATP levels compared to WEGG, particularly after 72 h of incubation. As the source of energy at the cellular level, ATP in the GFIPO group might have been consumed to a larger extent than in the WEGG group for energy, in the processes including lipid transport and fat accumulation, leading to the decreased ATP levels in the PCLSs. However, ATP levels remained stable throughout the incubation period with GFIPO, with an average ATP content of 4 pmol/µg protein (Figure 1C), which could be considered as viable according to our previous studies [6,8,10]. 

Following these observations on PCLS viability, we further investigated gene expression using NGS to gain a deeper insight into the differences between WEGG and GFIPO.

#### 3.1.2. Principal Component Analysis (PCA)

PCA was employed to ascertain the impact of experimental variables, namely, the medium and incubation time, on the data variance. The resulting two-dimensional plot, which was derived from the first two principal components (PC1 and PC2), is depicted in Figure 1D. Samples from the 0 h time point, obtained immediately after slicing, were separated along PC1, indicating an effect of incubation on liver slices compared to untreated slices. Despite the intrinsic human variability noted among the 0 h samples, this heterogeneity did not appear to substantially affect the variation of samples after incubation. All samples from 24 h incubation were separated together at the bottom part of the plot, regardless of whether they were in the GFIPO or WEGG medium. For the 48 h time point, the samples remained distinct from other time points, and, notably, they were also separated by different medium used. Upon reaching 72 h and 96 h, a clear separation between GFIPO and WEGG was observed. Overall, PCLSs can be differentiated based on both the medium and the duration of incubation time, suggesting that these two parameters could be the principal determinants influencing the effects observed in the PCLSs culture over time.

In assessing how PCLSs change with long-term incubation, we performed overrepresentation analysis on dysregulated genes in WEGG compared to the 0 h groups. It showed an increase in pro-inflammatory and pro-fibrotic processes (Appendix A) and a decrease in metabolic processes upon incubation (Appendix A). Therefore, our primary focus was to compare the influence of GFIPO on PCLSs to that of WEGG (control group) at each time point, aiming to assess if GFIPO would aggravate tissue dysfunction, inflammation, and fibrotic response in the context of MASLD progression.

#### 3.1.3. Total Number of Differentially Expressed Genes 

Next, we analyzed differentially expressed genes (DEGs) under various experimental conditions as shown in volcano plots (Figure 2A). At 24 h, we observed a relatively modest number of DEGs, comprising 59 induced genes and 33 repressed genes. However, a notable elevation in the number of DEGs was evident at 48 h, with 149 genes up-regulated and 159 genes down-regulated. This observation suggests that the influence of GFIPO may commence at around 48 h. Remarkably, at both 72 h and 96 h, we identified a substantial number of DEGs induced by GFIPO (175 induced and 507 repressed at 72 h, 164 induced and 435 repressed at 96 h). These findings align with our PCA analysis, reinforcing the notion that the impact of GFIPO builds up over time. Notably, our analysis revealed a larger number of down-regulated genes compared to up-regulated genes. This suggests that the influence of GFIPO may primarily entail the down-regulation of specific biological pathways.

#### 3.1.4. Overlapping Differentially Expressed Genes

To examine transcriptional alterations within each comparison, we conducted a cross-referencing analysis of the DEGs to identify shared genes across different time points. Figure 2B,C illustrate a relatively limited overlap in DEGs shared in all time points. Up-regulated gene sets have relatively more in common (19 out of 323 up-regulated genes), while down-regulated genes only share 7 out of 662 genes. When assessing each time point, we observed a progressive increase in the number of uniquely up-regulated DEGs over time, while this trend reversed for down-regulated genes starting at 72 h. Considering that the total amount of both up-regulated genes and down-regulated genes peaked at 72 h compared to other time points, the rising number of uniquely up-regulated DEGs after 96 h indicates a divergence in transcriptional profiles at this time point compared to 72 h. Additionally, our analysis suggests that the primary change in up-regulated genes occurs between 48 and 72 h, while predominantly down-regulated genes are found at 72 h–96 h (Figure 2D,E).

#### 3.1.5. Top 10 Significantly Differentially Expressed Genes

Next, we aimed to pinpoint the most significantly altered genes at each time point. The column charts in Figure 2F illustrate the top 10 up-regulated and down-regulated genes in GFIPO compared to WEGG, with the selection criteria based on thresholds (baseMean > 50, *p*.adj < 0.05, and log2FoldChange > 1). These genes were ranked according to their absolute log2FoldChange values. Appendix A provides additional information, including ensemble gene IDs and functions of encoded proteins, and a comprehensive list of all DEGs. The top 10 up-regulated genes exhibited substantial overlap across all time points and were implicated in diverse biological processes. These processes encompassed inflammatory response (*NOS2*), carbohydrate binding activity (*CRYBG2*), signaling transduction (*KSR2*), and endothelial cell activation (*ANKRD1*), which were predominantly observed in the early stages. Genes associated with cell cycle regulation (*S100A3*), cell survival (*FGF23*), cell adhesion (*IGFN1*), and cell migration (*ADAMTS5*) were prominent in later stages. On the other hand, down-regulated genes caused by GFIPO were primarily linked to metabolic processes (*DUOXA2*, *DUOX2*, *AQP7*, *STEAP4*, *G6PC1*, *ADH1C*, *PCK1*, *ADH1B*, *CFHR4*) and inflammation response (*C2CD4A*, *ADCY1*, *SIGLEC1*, *CLEC4G*, *CD5L*). Notably, a cluster of cytochrome P450-encoding genes (*CYP4A11*, *CYP3A4*, *CYP2A6*, *CYP2A7*) manifested alterations after 72 h of incubation. Similar trends were also observed in other studies where *CYP2E1*, *CYP2C19*, and *CYP2C8* showed down-regulation that followed the disease stage progression [25].

### 3.2. Characterization of PCLSs

#### 3.2.1. Genes Related to Fatty Acid Metabolism

Next, we specifically examined the DEGs and pathways related to fatty acid metabolism, comparing the effects of GFIPO and WEGG at different time points. Figure 3 illustrates the up-regulated (in red) and down-regulated (in green) DEGs induced by GFIPO in comparison to WEGG over time (A–D). We focused on four processes known to contribute to TG accumulation: fatty acid uptake, de novo lipogenesis (DNL), TG synthesis, and very low-density lipoprotein (VLDL) secretion [26]. At 24 h (Figure 3A), GFIPO up-regulated the expression of the *GOT2* gene, which might lead to increased fatty acid uptake. It also up-regulated *ACSL5*, *DGAT1*, *DGAT2*, and *PNPLA3*, implying the promoted TG synthesis, leading to the accumulation of lipid droplets and lipid secretion. At 48 h (Figure 3B), some genes associated with TG accumulation and secretion, such as *APOB*, *CIDEC*, and *PLIN5*, started to be down-regulated. A lower production of ApoB100 was observed in the GFIPO medium compared to the corresponding WEGG medium, although the down-regulation started from 24 h (Appendix A). However, the expression of *ACLY*, involved in DNL, was up-regulated, while *CPT1A*, which plays a role in mitochondrial β-oxidation, was down-regulated. At 72 h (Figure 3C), numerous genes related to fatty acid uptake, intracellular fatty acid transportation, TG synthesis, and TG secretion were down-regulated. Additionally, genes involved in fatty acid oxidation, such as *CPT1A* (mitochondrial β-oxidation) and *ACOX1* (peroxisomal β-oxidation), along with *CYP4A11* (α-oxidation), were also down-regulated. At 96 h, a similar pattern to 72 h was observed, although there were no dysregulated genes in mitochondrial β-oxidation. 

#### 3.2.2. Fatty Acid Metabolism Pathway

Next, we performed GSEA on hallmark pathways (Figure 4A–D). The results indicated that the fatty acid metabolism pathway was up-regulated at 24 h but subsequently down-regulated at 48 h and beyond. This aligns with the results shown in Figure 3A–D. The heatmaps in Figure 4A–D display the gene expression changes in the fatty acid metabolism pathway. The majority of genes involved in the breakdown of fatty acids, particularly through mitochondrial β-oxidation, showed up-regulation at 24 h. However, starting from 48 h and continuing to 96 h, there was a shift towards down-regulation, with more genes participating in this process. Additionally, several genes related to the metabolism of fatty acids were altered at 48 h, primarily showing down-regulation. Biological processes derived from Gene Ontology (GO) (Figure 5A) also indicated the up-regulation of fatty acid biosynthetic and metabolic process at 24 h and down-regulation at later time points. It has been shown that after the development of a more severe disease, lower levels of metabolic intermediates are needed to sustain growth compared with steatosis [25]. This may explain the down-regulation in metabolism. Meanwhile, fatty acid oxidation and fatty acid catabolic process were reduced as well from 48 h onwards. 

#### 3.2.3. Phenotype of Fat Accumulation on PCLSs 

To assess fat accumulation in slices, we measured TG content both in slices and in the medium. Figure 5B shows that TG in slices is significantly increased in GFIPO compared to WEGG, which indicates that GFIPO induces fat accumulation in slices more extensively after long-term incubation. The TG levels in PCLSs in WEGG remain constant. We found TG in the medium to be higher in GFIPO than in WEGG, although the levels decreased over time (Figure 5C). Figure 5D,E show the differences between GFIPO and WEGG at each time point. Overall, in slices, TG levels increased over time, but in the medium, they decreased. This difference may be due to the down-regulation of fatty acid transport (Figure 5A), which perhaps leads to suppressed VLDL export. H&E staining (Figure 5F) showed the presence of microvesicular steatosis in GFIPO after 48 h, and macrovesicular steatosis after 96 h, while WEGG did not induce steatosis. GFIPO also caused the down-regulation in genes involved in the metabolism of a very long-chain, long-chain, and unsaturated fatty acids after 72 h, and in short-chain fatty acid after 96 h at the gene level via GO analysis (Figure 5G).

### 3.3. Inflammatory Response in PCLSs by GFIPO Compared to WEGG

Next, we investigated the inflammatory response of PCLSs when exposed to GFIPO compared to WEGG. Hallmark pathway results showed that genes associated with inflammatory responses only started to be up-regulated after 48 h of incubation, in which TNFα signaling via nuclear factor kappa-light-chain-enhancer of activated B cells (NFκB) played a major role with 24, 25, and 21 genes involved at 48 h, 72 h, and 96 h, respectively (Figure 6A–D). Moreover, the increase in the pro-inflammatory cytokines involved (*TNFα*, *IL-6*, and *IL-1β*) was confirmed at the gene expression level through RT-qPCR (Figure 6E–G). The gene expressions of pro-inflammatory cytokines (*TNF*, *IL6*, *IL1β*) were up-regulated in slices upon treatment with GFIPO compared to that with WEGG at each time point, although the differences found were not statistically significant. In addition, the expression of *TNF* increased over time in slices incubated with GFIPO. WEGG led to similar trends. However, the expression of *IL-6* and *IL-1β* showed the opposite effect, as well as the production of TNFα measured in the medium (Figure 6H and Appendix A). The results of cytokine and chemokine secretion further demonstrated the pro-inflammatory effects of GFIPO on PCLSs (Figure 6H). GFIPO significantly increased the production of TNFα, IL6, IL8, interleukin-1 receptor antagonist protein (IL-1RA), and C-X-C motif chemokine ligand 10 (CXCL10) in the medium after 48 h of incubation. As mentioned above, TNFα showed a significant increase in GFIPO compared to WEGG at the later stages, but the overall trend was a decrease after 48 h. This may indicate that the contribution of long-term incubation to the further development of liver inflammation is limited, as no additional inflammatory cell infiltration can be induced in this ex vivo model. Additionally, we also observed an up-regulation in cell adhesion pathways and the intercellular adhesion molecule 1 (*ICAM1*)’s gene expression through NGS after incubation in GFIPO (Appendix A).

### 3.4. Development of Liver Fibrosis in PCLSs by GFIPO Compared to WEGG

To assess the development of liver fibrosis on PCLSs, we conducted GSEA and found that the TGFβ signaling pathway was up-regulated by GFIPO compared to WEGG at all time points (Figure 7A–D). the reactome pathway results demonstrated that GFIPO activated many processes participating in TGFβ signaling, especially at 72 h of incubation (Figure 7E). Regarding the mRNA expression of *COL1A1* and *ACTA2 (*Figure 7F,G), GFIPO led to an increasing trend, only having significant differences at 48 h in *ACTA2* expression. Previously, collagen genes, such as collagen type 1 alpha 1 chain (*COL1A1*), were shown to be consistently up-regulated with MASLD progression from steatosis to fibrosis [25]. Here, we also observed both an increasing trend and an increasing difference between GFIPO and WEGG. However, the production of pro-collagen 1a1 and TIMP1 protein was significantly up-regulated by GFIPO at the later stages (Figure 7H,I). The GSEA results also showed elevated focal adhesion and matrix metalloproteinases in WikiPathways, as well as ECM receptor interaction in Kyoto Encyclopedia of Genes and Genomes (KEGG) at the later stages (Appendix A). Meanwhile, PSR staining revealed the up-regulation of positive areas in GFIPO compared to WEGG, indicating increased ECM accumulation by GFIPO (Figure 7J).

## 4. Discussion

MASLD is increasingly recognized as a major health challenge, necessitating advanced research models to elucidate its pathology [27]. Studies have identified a multifaceted etiology behind hepatic steatosis, inflammation, and fibrosis in MASLD [28]. While animal studies are informative [29], human-specific pathophysiological responses require more accurate models. Therefore, human PCLSs emerge as a superior ex vivo alternative, offering a preserved tissue microenvironment and controlled experimental variables [6]. This study utilized healthy human PCLSs to simulate MASLD onset, showing that GFIPO—rich in sugars, insulin, and fatty acids—induces macrovesicular steatosis, inflammation, and ECM accumulation compared to controls. 

First, we evaluated the viability of human PCLSs for extended incubation as a model for MASLD. Our findings revealed that human PCLSs remained viable after 96 h in both GFIPO and the control medium, maintaining intact morphology and ATP levels. H&E staining indicated no significant toxicity from GFIPO. However, NGS highlighted significant gene-level differences between GFIPO and control conditions, affecting metabolic, inflammatory, and fibrogenic pathways. These findings support the use of human PCLSs as a valuable ex vivo model for prolonged studies under GFIPO conditions.

In this study, GFIPO significantly promoted TG accumulation in PCLSs, a key indicator of MASLD initiation. Notably, extended incubation with GFIPO also resulted in the formation of macrovesicles, as confirmed by H&E staining, a finding not previously reported in other ex vivo models.

Next, we investigated how GFIPO influences MASLD progression in human PCLSs with an initial focus on genes involved in fatty acid uptake and metabolism, which are key early MASLD processes [30]. GFIPO was found to increase the mRNA level of fatty acid transporters like *GOT2* (FA translocase/CD36) [31] at 24 h, potentially enhancing fatty acid binding to hepatocytes. The up-regulation of *CAV1* (Caveolin-1) may also play a role in fatty acid uptake, as it is essential for fatty acid translocase localization and function [32]. GO analysis suggested that fatty acid transport was initially up-regulated by GFIPO at 24 h, and then it started to be reduced after 72 h, probably due to cellular stress or damage. Consistent with the literature suggesting shifts from fatty acid uptake to synthesis as MASLD progresses [31], we noted increased DNL from 48 h onwards, implied by the elevated expression of *ACLY*, whose encoded protein is a key enzyme converting citrate to acetyl-CoA [33].

The assembly of TG molecules is a crucial process for storing and exporting fatty acids [26]. GFIPO initially increased the mRNA expression of GPAM and DGAT enzymes, key in TG synthesis, suggesting an early stimulation of lipid biosynthesis, which is also supported by the elevated GO pathways related to fatty acid biosynthesis. From 72 h, these biosynthetic pathways began to decrease. It aligns with a clinical trial where DGAT1 levels fell as liver disease progressed from MASH to cirrhosis [25]. Besides, in the same study, triacylglycerol (TAG) levels were uniformly elevated in patients with MASH and steatosis compared to healthy individuals but were lower in MASH than in steatosis alone. Specifically, MASH patients had reduced polyunsaturated fatty acids (PUFA)-TAGs but increased short and saturated fatty acyl chain-containing TAGs, with these levels decreasing further in cirrhosis [25]. Here, we also observed the down-regulation in pathways involved in very long-chain, long-chain, and unsaturated fatty acids after 72 h, and in short chain fatty acid after 96 h, which reflects the transition from steatosis to more severe liver conditions. Moreover, research in obese mice suggests that decreased TG synthesis would exacerbate liver damage in MASH, indicating that TG might help protect against liver injury [34]. Our NGS analysis implied a decrease in lipogenesis, which might link to liver damage and the progression of MASLD in our PCLSs model.

Furthermore, GFIPO decreased genes essential for VLDL assembly and secretion, alongside reduced lipid storage pathways, at 72 h and 96 h. This was mirrored by a decrease in ApoB100, a VLDL stabilizer [35], indicating suppressed lipid secretion. Malaguarnera et al. found that the diminished VLDL assembly and secretion during MASLD progression could result from lipotoxicity and cytokines activity affecting transcription factors for lipogenesis [36]. This may correlate with the lower lipogenesis and increased cytokine release observed in our study due to GFIPO. Interestingly, we also found a suppression in fatty acid oxidation implied by the decreased gene expression of key enzymes in mitochondrial β-oxidation (*CPT1A*), peroxisomal β-oxidation (*ACOX1*), and ω-oxidation or α-oxidation (*CYP4A11)*, as well as several reduced fatty acid oxidation pathways. Impaired β-oxidation leads to an abnormal TG accumulation and the development of MASLD [37]. Therefore, the suppression of lipid oxidation could potentially contribute to MASLD progression in PCLSs. 

Overall, our results suggest that GFIPO exerts a temporal effect on lipid metabolism. In the initial stages, GFIPO increases genes involved in fatty acid uptake and storage. With prolonged exposure, there is a down-regulation of genes related to fatty acid oxidation and secretion, while genes involved in DNL are up-regulated, which may lead to increased lipid accumulation. 

MASLD can progress from simple hepatic lipid accumulation to MASH, characterized by inflammation and fibrosis. Unlike current PCLSs models, which show only steatosis, lipid deposition, and lipotoxicity [10,11,12,38], human PCLSs under GFIPO after 96 h exhibited markers of liver inflammation, including up-regulated inflammatory genes (*TNFα*, *IL6*, and *IL1β*) and cytokines (TNFα, IL6, IL8), detectable up to 96 h. This was evidenced by RT-qPCR and cytokine assays, with NGS confirming the increase in inflammatory response over time. Noteworthy, IL1β, IL6, and IL8 showed earlier up-regulation than TNFα, suggesting that TNFα production might be activated by these pro-inflammatory cytokines, potentially through TNFα signaling via NFκB, as evidenced by the associated up-regulated genes from NGS. Additionally, MASH is also characterized by inflammatory cells’ infiltration [39]. Although immune cell recruitment is not directly observed in PCLSs, increased cell adhesion pathways and intercellular adhesion molecule 1 (*ICAM1)*’s gene expression (Appendix A) suggest a facilitated adhesion of immune cells to the endothelium, leading to inflammatory infiltration in MASH [40,41]. Thus, GFIPO-treated human PCLSs provide a model for studying the pro-inflammatory aspects of MASLD progression. 

In addition to liver inflammation, liver fibrosis is also a crucial phase in MASH development, largely driven by TGFβ signaling [42]. In our study, NGS showed that GFIPO up-regulated this pathway, specifically at 48 h and 72 h. In chronic liver injury or inflammation, TGFβ binds to its receptors, triggering a signaling cascade, for example through SMAD families, and promoting fibrosis [43]. Our study showed that TGFβ-activated SMAD2/SMAD3:SMAD4 heterotrimer signaling pathways were significantly enhanced under GFIPO on the gene level, consistent with elevated pro-fibrotic biomarkers like Pro-Collagen 1a1. PSR staining and TIMP1 production exhibited a similar trend. The TGFβ-stimulated TIMP1 production could inhibit matrix degradation enzymes, leading to ECM accumulation [43,44,45]. It has also been reported that up-regulated TGFβ activates hepatic stellate cells (HSCs), driving their transformation into collagen-producing myofibroblasts [46]. GFIPO also appeared to activate HSCs, as evidenced by increased *ACTA2* expression, suggesting an early sign of fibrogenesis. Additionally, focal adhesion and matrix metalloproteinases in WikiPathways, as well as ECM receptor interaction in KEGG, were all up-regulated in later stages (Appendix A), echoing gene patterns seen in cirrhosis [25]. Furthermore, a decline in metabolic genes, such as *MAT1A* and *GNMT*, could also be linked to severe MASLD (fibrosis stages 3–4) [47]. The inhibition of these genes leads to steatohepatitis, advanced liver fibrosis, and hepatocellular carcinoma in mouse models as well [47]. All of this supports the development of liver fibrosis induced by GFIPO at the later stages. 

The progression of MASLD is understood to be driven by the intricate interplay of inflammatory stress and lipid accumulation, aided by pro-inflammatory cytokines like interleukins and TGFβ [44]. This study suggests that the GFIPO-induced activation of TNFα and TGFβ pathways, coupled with the previously mentioned disruption in lipid metabolism, is implicated in the advancement of MASLD, characterized by liver inflammation and the development of fibrosis.

Hepatocellular carcinoma (HCC) often marks the terminal phase of MASLD with potential oncogenic signals emerging during the MASH stage [25]. Metabolic dysregulation, oxidative damage, chronic inflammation, and a fibrotic environment in MASLD promote hepatocellular destruction and compensatory proliferation, increasing the risk of genetic aberrations [48,49]. Our study indicated heightened cell growth, evidenced by increased pathways associated with cell cycle (Appendix A). Additionally, a persistent inflammatory status associated with the increased release of tumor necrosis factors could predispose to carcinogenesis specifically related to steatosis and steatohepatitis [50]. Moreover, fibrosis probably caused a poor oxygen exchange, leading to the increase in hypoxia (Appendix A). Hypoxia might stimulate tumor angiogenesis via the increased gene expression of vascular endothelial growth factor (VEGF) and hypoxia-inducible factor (HIF)-1 (Appendix A), which are identified as potential early indicators of HCC [51]. However, these results could also be triggered by inflammation or liver injury. Therefore, future studies could focus on genetic/epigenetic alterations to further elucidate the progression to HCC.

## 5. Conclusions

Our study demonstrates that human PCLSs treated with GFIPO for up to 96 h are a viable model for replicating the patient experience of MASLD, effectively simulating its progression, including MASH and fibrosis—stages that have been challenging to study in short-term PCLSs models. The use of NGS in our methodology sheds light on the complex processes of MASLD, allowing for a detailed exploration of its progression. To achieve an even more comprehensive pro-inflammatory environment, additional methods such as introducing pro-inflammatory stimuli or co-culturing with inflammatory cells could be employed. Diseased liver tissues from MASH cirrhosis patients or animal models could also be utilized for elongated incubation.

## Figures and Tables

**Figure 1 nutrients-16-00626-f001:**
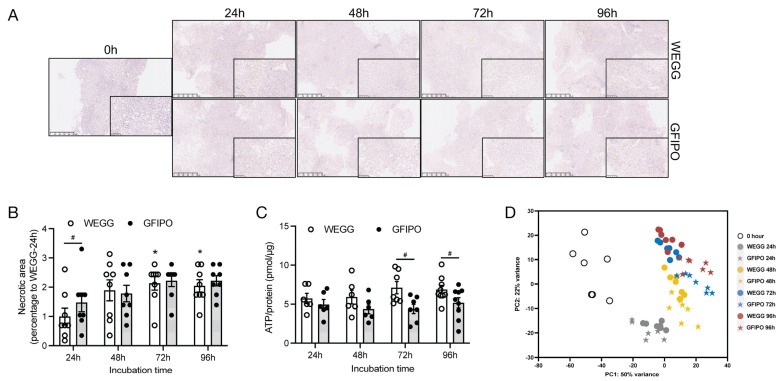
Initial characterization of human PCLSs, cultured for up to 96 h in WEGG or GFIPO. (**A**) Representative H&E staining images of PCLSs after long-term incubation in WEGG and GFIPO. Scale bar = 500 µm; inset: 100 µm. (**B**) Presence of necrotic areas compared to WEGG 24 h in H&E staining. (**C**) ATP/protein content in PCLSs after up to 96 h incubation in WEGG or GFIPO. (**D**) PCA showing the first two principal components. Each symbol corresponds to an individual patient sample composed of 3 slices, with a total of 7 patients (*n* = 7) assessed in each group. Data are presented as mean ± SEM. (#) denotes statistical differences between GFIPO and WEGG at each time point, while (*) denotes statistical differences in GFIPO or WEGG compared to their corresponding 24 h; *^(#)^
*p* < 0.05.

**Figure 2 nutrients-16-00626-f002:**
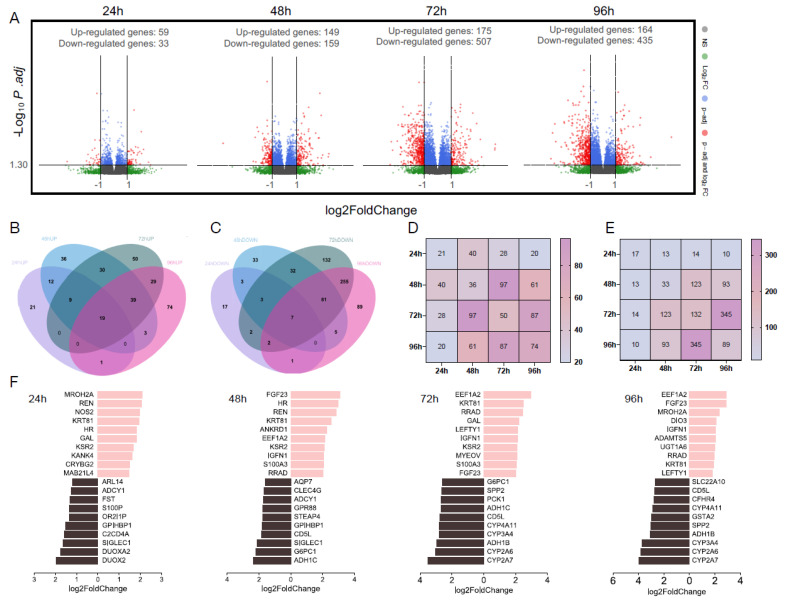
NGS of PCLSs, cultured for up to 96 h in WEGG or GFIPO. (**A**) Volcano plots representing the differentially expressed genes (red), with a threshold of log2FoldChange > 1 on the *x*-axis and *p*.adj < 0.05 (−Log10*p*.adj > 1.30) on the *y*-axis, comparing GFIPO and WEGG at each time point. WEGG serves as the control group. (**B**,**C**) Venn diagrams illustrating the distribution of up-regulated genes (**B**) and down-regulated genes (**C**) across all time points. (**D**,**E**) Heatmap showing the number of overlapped up-regulated (**D**) and down-regulated (**E**) DEGs between two timepoints; overlap of the same timepoint stands for the unique DEGs amount as shown in the Venn diagrams. (**F**) The top 10 up-regulated genes (highlighted in pink, right) and down-regulated genes (highlighted in black, left) at each time point.

**Figure 3 nutrients-16-00626-f003:**
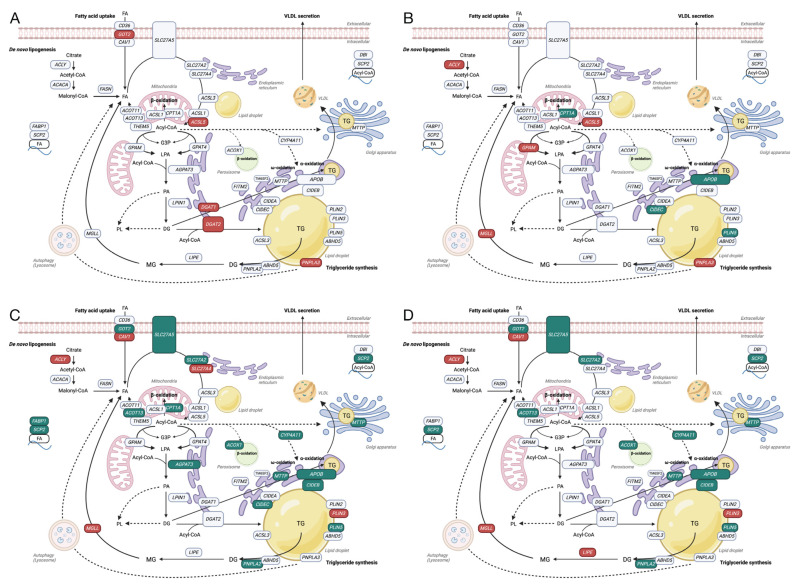
Gene regulation in fatty acid metabolism pathways by GFIPO compared to WEGG. (**A**–**D**) Up-regulated (in red) or down-regulated (in green) DEGs associated with fatty acid metabolism induced by GFIPO compared to WEGG at 24 h (**A**), 48 h (**B**), 72 h (**C**), and 96 h (**D**) of incubation.

**Figure 4 nutrients-16-00626-f004:**
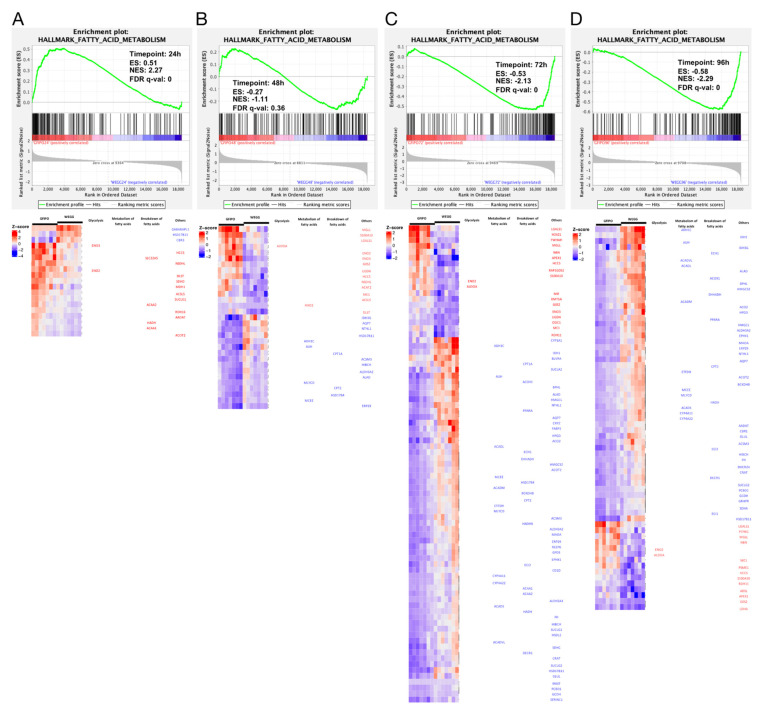
Gene regulation in fatty acid metabolism pathways by GFIPO compared to WEGG as identified by Gene Set Enrichment Analysis (GSEA). (**A**–**D**) GSEA plots of the hallmark fatty acid metabolism pathway at each time point: (**A**) 24 h, (**B**) 48 h, (**C**) 72 h, and (**D**) 96 h; below are significantly dysregulated genes in this pathway.

**Figure 5 nutrients-16-00626-f005:**
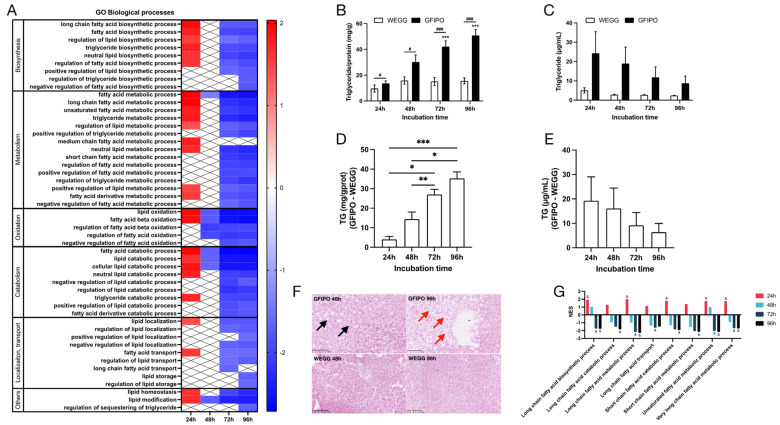
Fatty acid metabolism pathways by GFIPO compared to WEGG as identified by Gene Set Enrichment Analysis (GSEA) and phenotype of triglyceride regulation. (**A**) Altered biological processes associated with lipid, fatty acid, and TG metabolisms (nominal *p*. value < 0.01) at each time point, ranked by the normalized enrichment score (NES) value on the scale bar (NES > 0, up-regulated; NES < 0, down-regulated). (**B**,**C**) Increase in TG content relative to control at 0 h in PCLSs (**B**) or medium (**C**) after up to 96 h incubation in WEGG or GFIPO. D-E Differences in TG increase between GFIPO and WEGG in PCLSs (**D**) and medium (**E**). (**F**) Representative images of H&E staining on PCLSs (scale bar = 100 µM, black arrows indicate microvesicular steatosis, red arrows indicate macrovesicular steatosis). (**G**) Altered GO biological processes by GFIPO compared to WEGG in fatty acid with different chain lengths (& indicates significantly changed compared to the corresponding WEGG). Data are presented as mean ± SEM. (#) denotes statistical differences between GFIPO and WEGG at each time point, while (*) denotes statistical differences in GFIPO or WEGG compared to their corresponding 24 h; *^(#)^
*p* < 0.05, ** *p* < 0.01, ***^(###)^
*p* < 0.001.

**Figure 6 nutrients-16-00626-f006:**
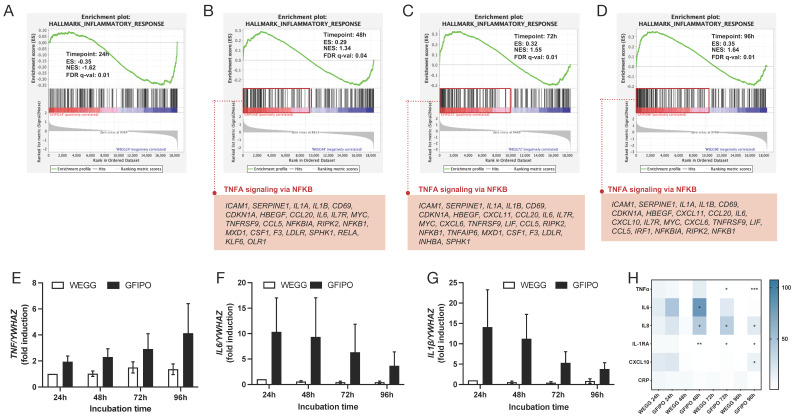
Gene regulation in inflammatory response of GFIPO compared to WEGG in PCLSs. (**A**–**D**) GSEA plots of hallmark inflammatory response pathway at each time point: (**A**) 24 h, (**B**) 48 h, (**C**) 72 h, and (**D**) 96 h; below are up-regulated genes involved in TNFα signaling via NFκB pathway. (**E**–**G**) mRNA expression of inflammatory biomarkers (*TNF*, *IL6*, *IL1β*) in PCLSs after up to 96 h of incubation. (**H**) Secretion of inflammatory cytokines by PCLSs after up to 96 h of incubation (100% to WEGG 24 h). Data are presented as mean ± SEM, (*) denotes statistical differences between GFIPO and WEGG at each time point; * *p* < 0.05, ** *p* < 0.01, *** *p* < 0.001.

**Figure 7 nutrients-16-00626-f007:**
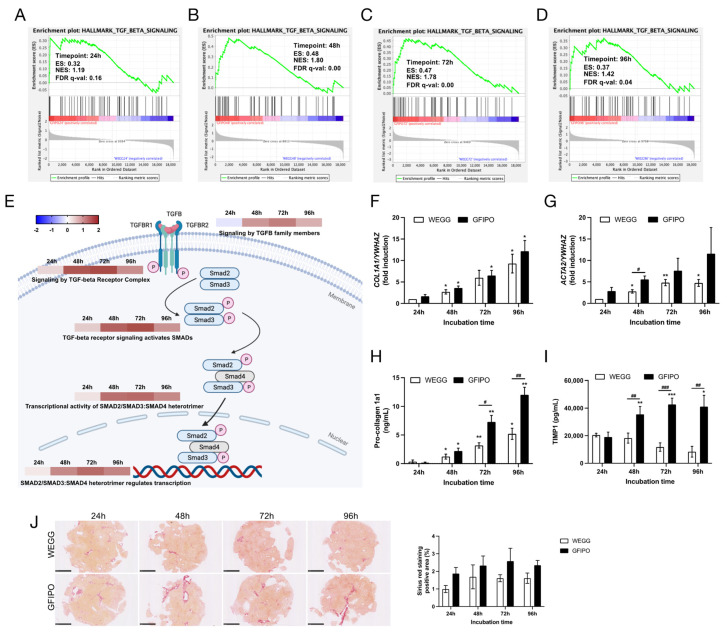
Development of liver fibrosis in PCLSs by GFIPO compared to WEGG. (**A**–**D**) GSEA plots of hallmark TGFβ signaling pathway at each time point: (**A**) 24 h, (**B**) 48 h, (**C**) 72 h, and (**D**) 96 h. (**E**) Dysregulation of reactome pathways (“Signaling by TGFβ family members”, “Signaling by TGFβ Receptor Complex”, “TGFβ receptor signaling activates SMADs”, “Transcriptional activity of SMAD2/SMAD3:SMAD4 heterotrimer”, and “SMAD2/SMAD3:SMAD4 heterotrimer regulates transcription”) in TGFβ signaling, ranked by NES value on the scale bar (NES > 0, up-regulated; NES < 0, down-regulated). (**F**,**G**) mRNA expression of inflammatory biomarkers (*COL1A1*, *ACTA2*) in PCLSs after up to 96 h of incubation. (**H**,**I**) Secretion of pro-collagen 1a1 and TIMP1 from PCLSs after up to 96 h of incubation. (**J**) Representative images of Picro Sirius Red staining on PCLSs (scale bar = 1 mM) and quantitative analysis of positive areas (control to WEGG 24 h) using ImageJ. Data are presented as mean ± SEM. (#) denotes statistical differences between GFIPO and WEGG at each time point, while (*) denotes statistical differences in GFIPO or WEGG compared to their corresponding 24 h; *^(#)^
*p* < 0.05, **^(##)^
*p* < 0.01, ***^(###)^
*p* < 0.001.

## Data Availability

The raw data supporting the conclusions of this article will be made available by the authors on request.

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
