# Peer review of "Metabolic Dysfunction-Associated Steatotic Liver Disease in a Dish: Human Precision-Cut Liver Slices as a Platform for Drug Screening and Interventions"

_nutrients, 2024, doi:10.3390/nu16050626_

Round 1

Reviewer 1 Report

Comments and Suggestions for Authors

This is an elegant in vitrostudy investigating time-dependent genetic, morphological and metabolic modifications observed in human precision-cut liver slices (PCLSs) cultured for 96 hours in a medium enriched with sugar, high fatty acids and insulin to induce MASLD. I have no remarks on methodological issues that seem to be perfectly done. However, I have sense of unease that deserves notice in discussion. It is clear that all experimental studies should be more or less reflect clinical settings. 

It is known that in MASLD patients the progression of liver disease is slow. Increase by one degree of fibrosis takes 7-14 years. How it is possible that the whole natural history of MASLD could be compressed to 96 hours. Do levels of fatty acids, insulin or sugar corresponded to those encountered in patients with metabolic syndrome?   

Reviewer 2 Report

Comments and Suggestions for Authors

The current paper describes the treatment of Precision cut liver slices with an updated protocol compared to other published/established by the same group. The aim is to mimic key characteristics of MASLD to create a translational ex vivo model based on human tissue. 

Other studies, including from the authors, have previously shown the potential of PCLS as an ex vivo model to study MASLD, but they succeeded in inducing mainly steatosis and TG accumulation. However, in previous papers the inflammation was not replicated lacking a key component of the human disease. 

Focus of this study was to prolong the treatment and to replicated more severe stages of MASLD not limited to fat deposition.

Fig.1 I agree with the authors about the limitations of measuring ATP as indicator of viability, therefore I would suggest adding another assay. Assays like LDH or cytokeratin 18 (M65, M30) or AST/ALT to evaluate cell death would help to rule out that changes in ATP may be due to its use in cell processes.

In fig 1D is not clear what each symbol represents, is it a single patient and multiple slices or each symbol corresponds to an individual patient? These details should be specified in the legend. 

In addition, authors should comment about patient heterogeneity in the PCA.

Fig1A the HE is very faint, I would recommend uploading better pictures. (This could also be an issue related to the resolution of current images so editorial team must check it.)

Fig.6H: the representation of the results is not very clear, I would recommend showing the results as mean and Standard dev and indicate in how many supernatants the cytokines have been measured. 

Reviewer 3 Report

Comments and Suggestions for Authors

The manuscript by Li et al. describes the use of human precisition-cut liver slices as a model of MAFLD. The authors used a good set of methods to characterize the PCLS. The experiments appear to be well performed.

I have the following points:

1. IHC with alphaSMA would supplement the data on fibrosis.

2. It is unclear which genes are validated by qRT-PCR. At least key mRNAs should be validated.

3. RNAseq data should be accessible in a data base of in the supplement.

4. Table 1 is not necessary.

5. It should be clearly indicated how many independent NGS had been performed per condition.

6. Fig. 1A and 2A cannot be seen and thus not be reviewed.
